# A Comparative Follow-Up Study of Patients with Papillary Thyroid Carcinoma Associated or Not with Graves’ Disease

**DOI:** 10.3390/diagnostics12112801

**Published:** 2022-11-15

**Authors:** Andrea Marongiu, Susanna Nuvoli, Andrea De Vito, Maria Rondini, Angela Spanu, Giuseppe Madeddu

**Affiliations:** 1Unit of Nuclear Medicine, Department of Medical, Surgical and Experimental Sciences, University of Sassari, 07100 Sassari, Italy; 2Unit of Infectious Diseases, Department of Medical, Surgical and Experimental Sciences, University of Sassari, 07100 Sassari, Italy

**Keywords:** Graves’ disease, papillary thyroid carcinoma, aggressive carcinoma, risk factors, follow-up, neck lymph node, distant metastasis, 131I SPECT/CT, thyroglobulin

## Abstract

Whether papillary carcinoma (PC) behavior is more aggressive in Graves’ disease (GD) patients than PC cases without GD is controversial. We retrospectively enrolled 33 thyroidectomized PC/GD patients during long-term follow-up, 23/33 without risk factors at surgery, and 18/33 microcarcinomas; 312 PC euthyroid-matched patients without risk factors served as controls. A total of 14/33 (42.4%) PC/GD patients, 4 with and 10 without risk factors at diagnosis, 6 with microcarcinoma, underwent metastases during follow-up. In controls, metastases in 21/312 (6.7%) were ascertained. Considering 10/23 PC/GD patients and 21/312 controls without risk factors who developed metastases, univariate analysis showed that there was an increased risk of metastasis appearance for PC/GD cases (*p* < 0.001). Disease-free survival (DFS) was significantly (*p* < 0.0001, log-rank test) shorter in PC/GD patients than in controls. Significantly more elevated aggressiveness in 6/18 PC/GD patients with microcarcinoma than in controls was also ascertained with shorter DFS. Thus, in the present study, PC/GD had aggressive behavior during follow-up also when carcinoma characteristics were favorable and some cases were microcarcinomas. GD and non-GD patient comparison in the cases without risk factors at diagnosis showed an increased risk to develop metastases in GD during follow-up, suggesting that GD alone might be a tumor aggressiveness predictive factor in these cases.

## 1. Introduction

The association between differentiated thyroid carcinoma (DTC), particularly papillary carcinoma (PC), and Graves’ disease (GD), varies in incidence and is still indeterminate [1,2,3,4]. GD is due to the activation of thyroid-stimulating autoantibodies (TRAbs) that stimulate an overproduction of thyroid hormones. Moreover, the activation of the TSH receptor by TRAbs hypothetically promotes cell proliferation, carcinogenesis, and angiogenesis with a high risk of cancer. Some authors have reported a correlation between cancer relapses after thyroidectomy and high levels of TRAbs in PC/GD patients [5]. However, other authors did not show a correlation between increased TRAbs and the prevalence of DTC in GD cases. Anyway, this finding does not eliminate the possible role of TRAbs in the induction of DTC since they may affect the proliferation of thyroid cells independent of their titers [6]. An alternative mechanism can be due to an elevated serum insulin-like growth factor that seems to be potentially associated with a risk of thyroid cancer due to its mitogenic and antiapoptotic properties [7]. Moreover, it has been observed that both IGF-1 [8,9] and vascular endothelial growth factor (VEGF) [9] have been found elevated in Graves’ disease. IGF-1 would seem to stimulate VEGF expression that in turn can potentially stimulate endothelial cell proliferation and angiogenesis and thus play a key role in the pathophysiology of thyroid carcinoma. Furthermore, anti-TPO and anti-Tg titles and environmental and genetic factors could also contribute to the risk of cancer in GD patients [4]. However, the intimate mechanism by which an increased frequency and aggressiveness of cancer could be verified in patients with GD is not yet well known. Anyway, GD seems to represent an increased risk of thyroid cancer in the prevalence of classic papillary carcinoma, although other more aggressive variants, such as tall cell carcinomas, have been reported [10]. 

Moreover, it is a matter of controversy, with conflicting and inconclusive results, whether cancer’s behavior is more aggressive in the presence of these two associated conditions. The frequent presence of neck and distant metastases in patients with PC and GD has been described [11,12,13,14]. Moreover, it has been reported that the recurrences and metastases during follow-up are more frequent in patients with PC/GD than in euthyroid cases affected by PC without GD [15,16,17,18,19,20,21,22,23,24]. Moreover, increased disease-specific mortality in DTC/GD patients in respect of matched euthyroid control cases was also reported [25].

However, other studies have not confirmed these data and have not shown any difference in the outcome between GD and non-GD thyroid carcinomas [26,27,28,29,30,31,32], including the pediatric and young adult patients [33], thus hypothesizing that GD does not affect the prognosis of PC. 

Metastases from DTC after thyroidectomy and radioiodine ablation are usually identified using ^131^I whole-body scan (WBS) [34,35,36] and the more recent SPECT/CT before [37,38,39,40] and after ablation [41,42,43,44,45,46,47], as well as during long-term follow-up [48,49,50,51]. Together with nuclear medicine procedures, radiologic exams and sequential serum thyroglobulin assays were performed.

The present study aimed to evaluate a group of patients with PC/GD in a long-term follow-up after total thyroidectomy and radioiodine ablation with and without risk factors at the surgery of primary cancer, trying to clarify whether GD may contribute to a less favorable progression of PC. For this purpose, we have compared the results obtained in PC/GD patients with those of a group of matched PC controls without GD. 

## 2. Materials and Methods

### 2.1. Patients

A total of 33 consecutive patients with PC/GD, 28 classic variants and 5 follicular variants, were retrospectively enrolled during the follow-up after total thyroidectomy and ablation with ^131^I. Their characteristics in the surgery of primary tumors are reported in Table 1. 

The diagnosis of Graves’ disease was made by clinical exams and laboratory thyroid function tests, and after diagnosis, all 33 patients were followed on pharmacological therapy (methimazole in all cases) for about 2 years. All PC/GD patients were in a state of clinical and biohumoral euthyroidism at the surgery after a long treatment with methimazole and the patients were no longer on treatment at the time of surgery. The patients were submitted to total thyroidectomy. The thyroidectomies of PC/GD patients were performed over a period of ten years, from 2006 to 2016.

The thyroidectomies were accompanied by central compartment lymph node dissection in 87.8% and by neck lateral lymph node dissection in 24.2% of cases. These latter were removed when metastases had been ascertained before surgery by imaging and biopsy or also if revealed during the operation. Total thyroidectomy was recommended to GD patients, according to their preference, when medical therapy for hyperthyroidism condition, methimazole in all cases of the present study, was not definitive for the cure of the disease because in the past some patients had experienced recurrences of the condition of thyroid hyperfunction and a surgical intervention was planned in agreement with the patients. However, the latter were euthyroid at the time of surgery, as mentioned above. Moreover, when the side effects of antithyroid drugs appeared, surgery was recommended. Further causes included the appearance of recurrent thyrotoxicosis or the presence of nodules in diffuse hyperplasia of the thyroid gland or large goiter found to be suspect of malignancy at neck ultrasound and nuclear medicine procedures or when there were difficulties to interpret fine-needle aspiration biopsy (FNAB) findings. 

The patients were aged <55 years in 26 cases and ≥55 in 7 cases.

A total of 15 patients were males, and 18 were females. The size of carcinoma was ≤10 mm (microcarcinomas) in 18 cases and >10 mm in 15 cases, and only in 2 cases of the latter the size was >20 mm. In 15 cases, the carcinomas resulted as incidental findings, all microcarcinomas, in diffuse thyroid hyperplasia (DTH) or a multinodular goiter (MNG) that were only identified during surgery or in post-operative histological section examination. In 10/33 patients, risk factors were ascertained, such as minimal extrathyroid extension (mETE) in three cases and laterocervical (LTC) metastatic lymph nodes (LN-N1b) in two cases and multifocality/multicentricity in six cases, one of the latter with also mETE, as above. In 23/33 cases, no risk factors were ascertained at the surgery. In six patients, exophthalmos was present. Only in 15 cases, all of them followed in our center, the result of TRAbs assay at the surgery of the primary tumor was available with positive values, while in the other 18 patients who came from other peripheral centers, TRAbs assay was not available. Moreover, TRAbs were not measured in the present series during follow-up. Following the classification of the European Thyroid Cancer Taskforce [52], the patients were classified as being at very low risk (VL-unifocal T1 (≤1 cm) N0M0 and no extension beyond the thyroid capsule), at low risk (L-T1 (>1 cm) N0M0 or T2 N0M0, or multifocal T1N0M0), or high risk (H-any T3 and T4 or any T, N1 or any M1). Thus, the 33 GD patients were classified as VL (T1aN0M0) in 12 cases, L (multifocal/multicentric T1aN0M0 or T1b/T2N0M0) in 19 cases, and H (T1bN1bM0) in 2 cases.

As reported in the same Table 1, 312 patients with PC without GD, 264 classic variants, and 48 follicular variants, from our database of thyroid cancer, were also enrolled as controls matched for sex, age, and tumor size submitted to total thyroidectomy and radioiodine ablation in the same period as GD patients. All control patients had no risk factors at surgery (such as ETE, mETE, multifocality/multicentricity, and neck and distant metastases). At risk stratification, 148 patients were L and 164 VL. Indication for the surgery was the presence of single or multiple euthyroid nodules in goiter in part suspected of cancer at an ultrasound and or FNAB findings. No thyroid autoimmune disease in the histological examination and/or in the serum of control patients was detected.

### 2.2. Methods

Both PC/GD and control PC patients were monitored in a long-term follow-up after thyroidectomy and radioiodine ablation with a mean period of 137.5 ± 75.9 months for PC/GD cases and 130 ± 41.8 months for PC without GD cases, *p* = 0.374. The patients were sequentially submitted to clinical examination, serum thyroglobulin assay and antithyroglobulin antibodies (AbTg), ultrasound imaging, ^131^I whole-body scanning (WBS), and SPECT/CT, 24–48 and 72 h after 185 MBq radioiodine diagnostic dose using a hybrid dual-head gamma camera. Before radioisotopic exams, all patients had a preparation with a low-iodine diet for 2 weeks and up to 24 h after the scan, avoiding iodine-containing medications. In the course of radioiodine ablation, ^131^I WBS and ^131^I SPECT/CT were performed in both 33 PC/GD patients and 312 control patients in hypothyroidism after the L-thyroxine withdrawal. However, during the follow-up period to which the present study refers, the nuclear medicine procedures were carried out in 23/33 patients of PC/GD patients (69.7%) in hypothyroidism and 10 cases (30.3%) after rh-TSH stimulation; moreover, 198/312 control patients (63.5%) underwent nuclear medicine procedures in hypothyroidism while 114/312 cases (36.5%) after rh-TSH stimulation. The patients submitted to rh-TSH stimulation were older and/or with heart problems. Furthermore, the stimulation mode between GD and control patients did not show any difference. According to standard procedures, all patients underwent laboratory tests, such as the assay of serum TSH with levels always over 50 μU/mL and serum thyroglobulin levels (cut-off value 0.2 ng/mL) and antithyroglobulin antibodies (Ab-Tg cut-off: 100 IU/mL). Serum thyroglobulin and AbTg were measured by using chemiluminescent immunoassay methods. The detection limit of thyroglobulin assay is 0.1 ng/mL. The cut-off for thyroglobulin positivity was considered <0.2 ng/mL during suppressive therapy and <1 ng/mL after stimulation.

### 2.3. Data Analyses

In our Nuclear Medicine Center of the University Hospital, the site of the present study, all radioisotopic instrumental examinations were performed. Four nuclear medicine physicians (A.M., S.N., A.S., and G.M.), who were aware of the reason for the exams but unaware of the results of the other previous investigations, independently analyzed ^131^I-WBS and SPECT/CT. The physicians are very experienced in both these procedures in DTC patients. Interobserver variability was very low, and disagreements were resolved by consensus. 

SPECT/CT findings were classified as normal in the presence of tracer distribution in normal tissue and physiologic structure and positive in the presence of neoplastic foci. ^131^I-uptakes were considered neoplastic in the presence of focal areas of increased accumulation that are not compatible with physiological storage locations. Moreover, WBS findings were classified as unclear when it was not easy to give a precise anatomic localization and/or characterization. SPECT/CT represented an added value compared to WBS when it provided better identification, correct anatomic localization and interpretation, and precise differentiation between neoplastic foci and normal tissue or physiologic uptakes. 

Metastasis status was identified by SPECT/CT and WBS, whose results have been compared with each other and confirmed by histology or by clinical and radiologic exams and thyroglobulin changes during the long-term follow-up when histology was unavailable. The prevalence of metastases in the follow-up of the PC/GD patients with that of PC cases without GD was also compared.

### 2.4. Statistics

The Shapiro–Wilk test was used to assess the normality of quantitative data. Quantitative variables were summarized with mean ± standard deviation (SD) or medians and 25–75° percentiles (IQR), whereas qualitative ones were by absolute and relative (percentages) frequencies. Subgroup differences of quantitative variables were evaluated by the Mann–Whitney test or Student t-test, as appropriate. Pearson chi-square or Fisher exact tests were used to assess differences for qualitative variables. Categorical variables were evaluated with the Fisher chi-squared test. A logistic regression analysis was performed to test the association between the collected variable and the risk of metastasis. The multivariate logistic regression included independent variables resulting in a *p* < 0.10 in the univariate analysis. Metastases were considered a dependent variable. The significance level was defined as *p* < 0.05.

Kaplan–Meier curves were plotted to assess 10-years disease-free survival visually, using the log-rank test to assess the statistical difference between GD and non-GD patients. McNemar test was used to assess the existence of disagreement between WBS and SPECT/CT data. A two-tailed *p* < 0.05 was considered statistically significant. All statistical analyses were performed with STATA version 16.1 (StataCorp. LLC, College Station, TX, USA).

## 3. Results

As previously illustrated in Table 1, where the characteristics of the patients (PC/GD and controls) has been reported, it was ascertained that no statistical difference existed in age (*p* = 0.269) and sex (*p* = 0.09) of patients, as well as in histology (*p* = 0.972) and in diameter of carcinomas (*p* = 0.828).

Table 2 shows that 14/33 (42.4%) patients with PC/GD (11 with PC classic variant and 3 with PC follicular variant, 6 with microcarcinoma) underwent metastases during follow-up (Group 1, Cases 1–14). A total of 10/14 of the latter Group 1 patients (Cases 1–10), 5 of them with microcarcinoma, did not show risk factors at surgery, while the remaining 4/14 cases (Cases 11–14), 1 with microcarcinoma, had risk factors. Moreover, 19/33 (57.6%) patients (17 PC with classic variant and 2 PC with follicular variant, 12 with microcarcinoma) did not develop metastases (Group 2, Cases 15–33).

Globally, in the 10 Group 1 patients with the absence of risk factors, ^131^I-SPECT/CT detected 13 neck LN metastases, 8 laterocervical (LTC), 4 submandibular (SM), and 1 paratracheal (PT); 2 of these were unclear, 1 wrongly classified as residue (Case 6), and 10 occult at WBS. Thyroglobulin was undetectable in 7 cases (one with AbTg positive—Case 9) and <2.5 ng/mL in 3 cases. Two of the patients, Case 5 and Case 9, are illustrated in Figure 1 and Figure 2, respectively. 

Of the remaining 4/14 Group 1 cases (Cases 11–14), 2/4 of patients with unifocal carcinoma and neck LN metastases at surgery developed a new neck LN metastasis (1 LTC) in one case, positive at SPECT/CT but occult at WBS, with undetectable thyroglobulin. However, the other patient developed five lung metastases during the follow-up positive at both SPECT/CT and WBS, and with thyroglobulin levels of 33 ng/mL. This latter patient was submitted to four radioiodine therapeutic doses with a partial response: only one of the five metastases localized in the perihilar region of the right lung persisted unmodified despite therapy with an overtime increase of the thyroglobulin levels. This metastasis resulted also positive at ^18^FDG PET/CT with a high SUV (14.50), and the patient was enrolled for performing target therapy with tyrosine kinase receptor inhibitors (lenvatinib), and the first response was partial with a reduction of metastasis size and thyroglobulin levels (0.84 ng/mL). The patient is still alive at the moment of the last follow-up visit. The other 2/4 patients, 1 with unifocal microcarcinoma and 1 with multifocal carcinoma, had mETE at surgery, and SPECT/CT in the follow-up detected two neck LN metastases (1 LTC, 1 SM) occult at WBS. Thyroglobulin in these two latter patients was <2.5 ng/mL.

In the 19 cases of Group 2 (Cases 15–33), 13/19 patients (Cases 15–27), 7 with microcarcinoma, had no risk factors at surgery and resulted negative for metastases at both SPECT/CT and WBS during the follow-up with undetectable thyroglobulin and absence of AbTg. Of the other 6/19 patients (Cases 28–33), one of whom with mETE and the remaining five cases with multicentric microcarcinoma at surgery, resulted concordantly negative for metastases at SPECT/CT and WBS during follow-up with undetectable thyroglobulin and absence of AbTg.

As illustrated in Table 3, of 312 PC control cases, all of them without risk factors at surgery, 21 patients (8 VL, 13 L), 8 of whom with microcarcinoma and only one >20 mm, 15 with PC classic variant and 6 with PC follicular variant, developed metastases during follow-up. Twenty-four neck LN metastases (11 LTC, 5 SM, 6 PT, 2 supraclavicular (SC)), two mediastinum (M), and one pelvic LN (PV) metastases were identified by SPECT/CT; 14/27 of these were occult and 13/27 unclear at WBS. Thyroglobulin was >10 ng/mL in one case, between 5 and 10 in 4 cases, between 2.5 and 5 in 5 cases, <2.5 in 3 cases, and undetectable in 8 cases; AbTg were absent in all cases.

The results of ^131^I WBS and SPECT/CT in identifying the metastases in the global casuistry, including PC/GD and PC euthyroid control patients, demonstrated that SPECT/CT obtained a significantly (*p* < 0.0001) higher performance than WBS, being able to evidence and characterize 48 metastases while WBS 21, most of the latter classified as unclear. 

Globally, the percentage of PC/GD patients who underwent metastases was 42.4% (14/33 patients), while in PC without GD patients was 6.7% (21/312); the difference was statistically significant (*p* < 0.001).

The mean age of PC/GD patients and of euthyroid control cases who developed metastases was not different (45.8 ± 11.9 and 45.6 ± 13.4 years, respectively, *p* = 0.962).

In the 14 PC/GD patients, metastasis appearance occurred in a shorter time in the 4/14 cases with risk factors at surgery (median 24 IQR 23.5–30 months) in respect of the 10/14 cases without risk factors (median 24 IQR 21–28 months). Moreover, in both of these groups, metastasis appearance occurred in a shorter time in respect of the 21 control patients without risk factors (median 28 IQR 24–37 months). However, the difference between the three groups was not statistically significant (*p* = 0.167). 

The disease-free survival (DFS) at 10 years’ PC/GD patients was significantly lower in respect of the patients without GD (56% vs. 93%, *p* < 0.0001), as reported in Figure 3. 

Comparing the 33 PC/GD patients, 30.3% of whom with risk factors at surgery, with the 312 control PC cases without GD and free of risk factors, analysis multivariate showed that PC/GD patients had a 10-fold higher risk of developing metastases than PC patients without GD in the follow-up (Table 4).

There have been also considered exclusively 23/33 of PC/GD patients all without risk factors, 43.5% (10/23) of whom developed metastases in the follow-up. These patients have been compared with the control PC patients without GD free of risk factors and in whom metastases developed in 6.7% of cases, as previously reported. Univariate analysis showed that there has been an increased risk of metastases appearance for GD patients during follow-up (odds ratio: 10.66 (95% CI 4.18–27.17) *p* < 0.001).

Furthermore, it was observed that there has been a higher incidence of microcarcinoma in PC/GD patients (18/33—54.5%) in respect of euthyroid PC controls (164/312–52.6%), but not significantly (*p* = 0.828). Among the 18/33 PC/GD patients with microcarcinoma, 12/18 cases (66.7%), 5 of whom were multicentric, and most of these identified as incidental findings at histology, had a favorable prognosis. However, 6/18 (33.3%) PC/GD cases, 5 of whom were ascertained incidentally at the surgery of primary tumor, developed metastases, as did 8/164 (4.9%) control patients with microcarcinoma. The difference was statistically significant (*p* < 0.0001). In particular, one of these six microcarcinomas in PC/GD patients had risk factors (mETE) with a size of 7 mm. Of the other five cases, all of these without risk factors at surgery, one had a tumor size <5 mm, while the remaining four cases had a size ranging from 5 to 8 mm. In these 6/18 PC/GD patients with microcarcinoma, metastases appeared in a shorter time (median 23.5 IQR 21–24 months) in respect of the 8/164 control patients (median 42 IQR 27–65.5 months) with a borderline statistical difference (*p* = 0.0519). 

Moreover, DFS in PC/GD patients with microcarcinoma (≤10 mm), was also significantly lower in respect of PC without GD patients (65% vs. 95%, *p* < 0.0001), as illustrated in Figure 4.

Of 15/33 PC/GD patients with carcinoma size >10 mm, 8/15 developed metastases (53.3%), 3 with risk factors, while 7/15, 1 with risk factors had a favorable prognosis (46.7%). Moreover, of 148/312 control patients without risk factors with carcinoma >10 mm, 13/148 cases (8.8%), developed metastases, while in 135/148 cases, the carcinomas were not aggressive (91.2%). The difference between these groups of patients was statistically significant (*p* < 0.0001).

In 8/15 PC/GD patients with carcinoma size >10 mm, who metastasized during follow-up, metastasis appeared with a median of 25 IQR 23–29 months, while in the 13/148 control patients, the median was 25 IQR 22–30 months, *p* = 0.962. 

DFS in PC/GD patients with carcinoma >10 mm was significantly lower than in PC without GD cases (43.2% vs. 91%, *p* < 0.0001), as shown in Figure 5.

Exophthalmos was present in six PC/GD cases at surgery, three of whom developed metastases.

None of the 33 PC/GD patients had a relapse of hyperthyroidism during the follow-up; for this reason, they are not treated with methimazole anymore.

## 4. Discussion

In the present, retrospective study, the ongoing debated problem of the relationship between GD and PC has been discussed, specifically the behavior of carcinoma associated with GD in the long-term follow-up after thyroidectomy and radioiodine ablation.

For this purpose, we enrolled a group of GD patients consecutively observed, in whom a PC was ascertained, this type of carcinoma representing the most frequent cancer associated with GD. The PC/GD patients had or did not have risk factors at the surgery of the primary tumor, such as multifocality, mETE, and neck and distant metastases. However, a group of euthyroid PC patients, who have been operated on in the same period of PC/GD and served as controls, was exempt from risk factors. Both PC/GD and euthyroid control patients had no statistically significant difference in age, sex, and tumor size and were monitored in a long-term follow-up. 

We observed a considerable, significant high rate of disease progression during the follow-up in PC/GD patients compared to euthyroid PC controls without GD, despite similar tumor characteristics. In most PC/GD cases, neck and distant metastases appeared in a shorter interval of time after thyroidectomy and radioiodine ablation in respect of controls, although the difference was not statistically significant (*p* = 0.167).

Moreover, the frequency of neck lymph node metastases was also higher in PC/GD patients in respect of matched euthyroid control patients during the follow-up, with PC/GD cases, but not controls, also developing distant metastases.

In particular, 4/14 PC/GD patients who developed metastases had risk factors at the surgery of the primary tumor, such as mETE, multifocality, and neck lymph node metastases, suggesting that GD may contribute to their onset during the follow-up together with the other risk factors. However, in the remaining 10/14 PC/GD cases without risk factors at the diagnosis and who underwent metastases, it may not be ruled out that GD alone could represent an independent predictive factor for the risk of metastasis with worsening disease prognosis and less favorable outcome. This result has been obtained by a univariate analysis comparing PC/GD and PC without GD patients, both groups being without risk factors at diagnosis.

These results seem to demonstrate that PC with associated GD was more aggressive than PC without GD even if none of the patients PC/GD died as a consequence of the tumor during the period of the study similar to what was observed in the euthyroid control patients.

The results obtained in this study regarding the aggressiveness of PC in GD patients were also reported by some authors but were in contrast with others; however, the intimate cause of these controversial data is still unclear.

In particular, some authors reported that the cumulative risk of recurrent/progressive distant metastases was approximately 3-fold higher in patients in PC/GD than in euthyroid DTC without GD patients [15,16]. Moreover, the same authors in another study [25] found 57.1% disease-free patients in DTC/GD group, compared to 87.1% in euthyroid DTC patients after a mean follow-up of 214 ± 18 and 187 ± 12 months, respectively. In the same study, the authors found that 33.3% (7/21) of GD patients and 10.0% (7/70) of non-GD patients were classified as stage IV on the TNM 7th edition. It has been reported that metastases occurred particularly in larger tumors [16,17], and in some cases, Graves’ disease was associated with an increased risk of persistent/recurrent disease only in tumors ≥1 cm when compared with euthyroid PC patients [23]. More increased cancer-specificity mortality in DTC patients with associated GD was also observed with distant metastases that represented the most frequent cause of tumor aggressiveness [25]. 

The results of the present study are in contrast with the data reported by some authors [26,30]. They did not find an increased aggressiveness of thyroid cancer in patients with GD during the follow-up when compared with euthyroid DTC subjects, either in large tumors or in microcarcinomas. Moreover, no difference in the persistence of the disease and cancer-related death was ascertained with an excellent prognosis and disease-free survival rate compared to those of patients with euthyroid status as observed by other authors [31].

In the present study, the highest number of PC/GD was represented by microcarcinomas (18/33—54.5%), similar to what has been observed by other authors [4,14,17,18,19,21,24,26,27,29,30,32,33] and especially the carcinomas incidentally found in the resected specimen [53].

The majority of small-size carcinoma in our series was classified as T1, also including those with mETE, according to AJCC Cancer Staging Manual 8th edition. Most patients (12/18) with microcarcinoma, most incidentally identified, and including 6/12 with risk factors at surgery, had an excellent outcome and longer free disease survival compared to age, sex, and tumor size-matched non-GD patients. Similar data were also evidenced in the majority of cases described by other authors [4,14,24,27,29,30], with no patients evidencing relapses at any time until the last control visit [25]. Thus, GD appeared not to affect the patient’s outcome. Similarly, in pediatric patients in whom incidental microcarcinoma is common, it has been reported no risk of tumor aggressiveness associated with small-size carcinoma (<1 cm) demonstrating a favorable long-term clinical course and an excellent prognosis compared with pediatric patients with DTC alone [33].

However, considering the remaining 6/18 PC/GD patients with microcarcinoma in the present study, all except one without risk factors at the surgery of primary tumor, it has been observed that they developed neck lymph node metastases during follow-up with thyroglobulin levels undetectable or very low. Moreover, the percentage of patients with metastases in PC/GD cases was higher in respect of euthyroid PC controls with microcarcinoma without risk factors and with metastatic lymph nodes, and DFS was significantly lower in PC/GD cases. These data obtained in the present study, in contrast with those of the other patients with microcarcinoma who had a favorable outcome and with the aforementioned data of the other authors, seem to suggest that GD, as the only risk factor, may have an impact on the metastasis appearance in these cases. Because of such results, it is considered necessary not to underestimate microcarcinomas in PC/GD patients during a long-term follow-up to better guarantee careful surveillance of affected patients, as also reported in previous studies in patients with thyroid papillary microcarcinomas associated or not with GD [50].

Moreover, the presence of multifocality/multicentricity was observed in 6/33 (18.2%) of PC/GD patients at the surgery of the primary tumor. Only one of the six cases had other risk factors, such as mETE. Except for the latter case, which developed neck metastases during the follow-up, the other five patients with multicentric microcarcinomas had a favorable outcome. These results are in agreement with the data reported also by other authors who observed that multifocality/multicentricity in their PC/GD series was not associated with disease recurrent/persistence during follow-up nor with a significant increase in the risk of DTC-related mortality [14,24]. However, in other studies on larger case series, multifocality, together with other risk factors, including a few cases with GD, was associated with metastasis appearance in the follow-up of PC patients [47,51].

The finding of exophthalmos in 18.2% of PC/GD series did not seem to affect the outcome of the GD patients since an identical percentage of cases developed metastases or, on the contrary, had a favorable prognosis. These results differ, at least partially, from those of other authors who found that the presence of clinical orbitopathy at baseline was associated with a decreased risk of events at the end of follow-up in DTC/GD patients [24].

As previously reported, all PC/GD patients were in a state of clinical and biohumoral euthyroidism at the surgery. Thus, the absence of an active Graves’ disease does not allow the exclusion of an eventual protective role of the florid inflammatory infiltrate, organized in the germinal center, onto carcinogenesis, as hypothesized and reported in other studies [54]. 

Moreover, none of the PC/GD patients of the present study relapsed into hyperthyroidism during the follow-up, unlike what was observed by other authors who reported relapse of the disease in some patients of their PC/GD series [30]. Thus, no PC/GD patients were treated with methimazole anymore.

The pathophysiologic mechanism of PC in GD cases by circulating TRAbs cannot be considered in the present study, since the assay of TRAbs is limited to a few cases only at the surgery of the primary tumor. Thus, their effect on tumor aggressiveness could not be investigated in the follow-up after thyroidectomy and radioiodine ablation.

Taking into account the diagnostic imaging procedure used in the present study, ^131^I-SPECT/CT has proven for several years to be a valid diagnostic procedure in the follow-up of patients with DTC. The procedure demonstrated its usefulness to identify metastases in both PC with associated GD and PC without GD of the present series also when thyroglobulin levels were undetectable or very low. SPECT/CT obtained a significantly higher performance than the traditional WBS method in respect of which the tomographic procedure ascertained a significantly more elevated number of metastatic lesions. Thus, routine use of SPECT/CT is suggested in the diagnostic protocol of papillary carcinoma with associated or not GD in the follow-up after thyroidectomy and radioiodine ablation.

However, some limitations due to the retrospective nature of this study should be examined, since some information may be lost. However, in our study, these limits did not seem to affect our analysis significantly. Anyway, the number of Graves’ patients is not elevated. 

Given the slow growth of PC, a metastatic lesion can appear negative at the first exam, while it can be evidenced in a late phase. Considering this aspect, a long follow-up is necessary since it is not excluded that some patients might develop metastases later. However, in the present study, the patients were followed for a long period, but it cannot be excluded that the follow-up is not long enough to detect when occult metastasis can become clinically evident even with undetectable thyroglobulin levels. 

The lack of histopathologic findings of some radioiodine-avid foci evidenced by SPECT/CT has been found because of the difficulty of reaching the potential site of the lesions. Only the data obtained during follow-up by clinical exams, thyroglobulin sequential variations, and radiologic and nuclear medicine imaging could validate the foci detected by SPECT/CT when histology was not available.

## 5. Conclusions

The results of this study seem to confirm that in patients affected by GD, an associated PC has a more aggressive behavior with a less favorable outcome in respect of euthyroid-matched PC cases, with the appearance of a higher percentage of metastases during follow-up and a shorter DFS. A more elevated aggressiveness than controls was also ascertained in some patients with microcarcinoma, unlike what was reported in other studies, and even when tumor characteristics were favorable. About that, the comparison between PC/GD patients of the present series with PC euthyroid controls, both of them with an absence of risk factors at the surgery of primary tumors such as mETE, multifocality, and neck and distant metastases showed that there was a significantly increased risk to develop metastatic lesions in GD patients during the follow-up. This result would seem to suggest that GD alone might be a predictive factor of tumor aggressiveness in these cases with worsening disease prognosis and with a less favorable outcome. However, a more elevated number of patients to confirm the results of the present study is necessary.

## Figures and Tables

**Figure 1 diagnostics-12-02801-f001:**
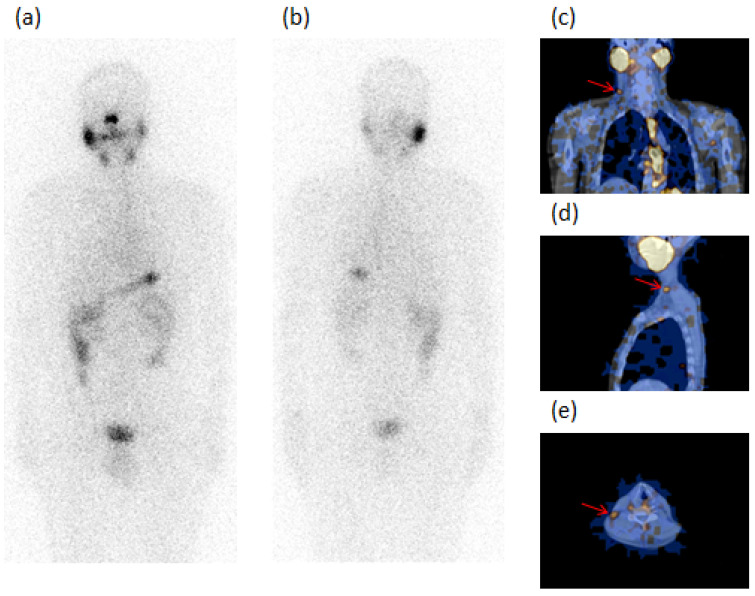
A 48-year-old male patient (Case 5, Table 2) submitted to total thyroidectomy and radioiodine ablation for papillary carcinoma (PC) classic variant, tumor diameter of 4 mm with associated Graves’ disease (GD). The figure refers to an image acquired after 24 months of follow-up. ^131^I-WBS, after a diagnostic dose of 185 MBq, in both anterior (**a**) and posterior (**b**) views did not evidence pathologic radioiodine-avid foci. SPECT/CT in coronal (**c**), sagittal (**d**), and transaxial (**e**) slides showed a radioiodine-avid focus in right laterocervical region of the neck (red arrow) which was classified as metastatic lymph node, confirmed at surgery. Thyroglobulin: undetectable. AbTg: absent.

**Figure 2 diagnostics-12-02801-f002:**
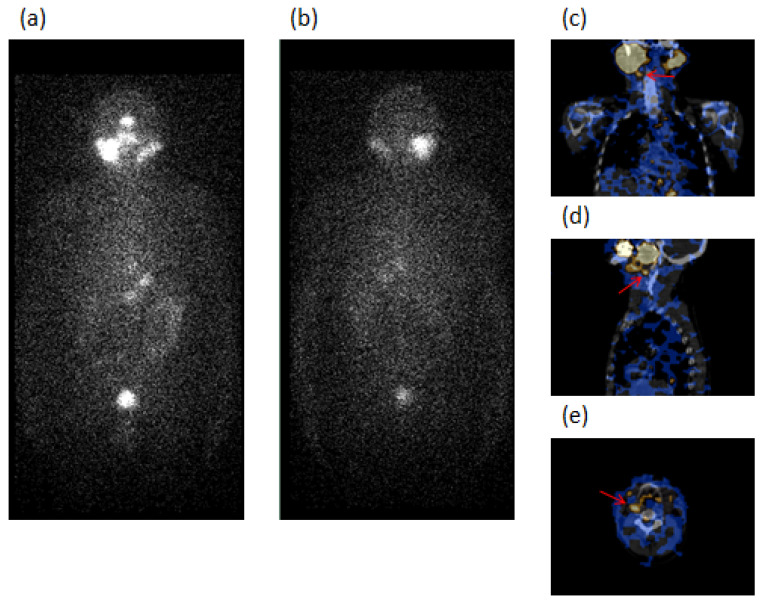
A 64-year-old male patient (Case 9, Table 2) submitted to total thyroidectomy and radioiodine ablation for papillary carcinoma (PC) classic variant, tumor diameter of 11 mm with associated Graves’ disease (GD). The figure refers to an image acquired after 22 months of follow-up. ^131^I-WBS, after a diagnostic dose of 185 MBq, in both anterior (**a**) and posterior (**b**) views excluded the presence of pathologic radioiodine-avid foci. SPECT/CT in coronal (**c**), sagittal (**d**), and transaxial (**e**) slides showed one radioiodine-avid focus in the posterior region to the submandibular right gland (red arrow). The focus was classified as metastatic lymph node and treated with radioiodine ablation. Thyroglobulin: undetectable. AbTg: 143 IU/mL.

**Figure 3 diagnostics-12-02801-f003:**
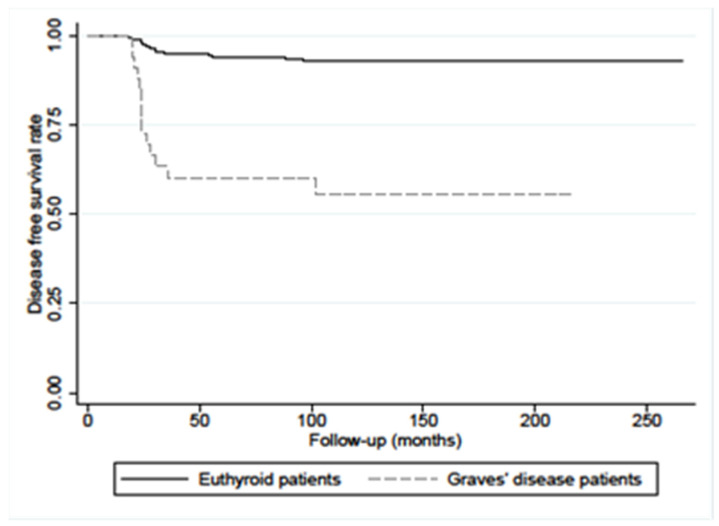
Disease-free survival (DFS) in PC/GD patients (n.33) and in matched euthyroid PC control patients (n.312) showed a statistical difference (*p* < 0.0001 calculated with log-rank test).

**Figure 4 diagnostics-12-02801-f004:**
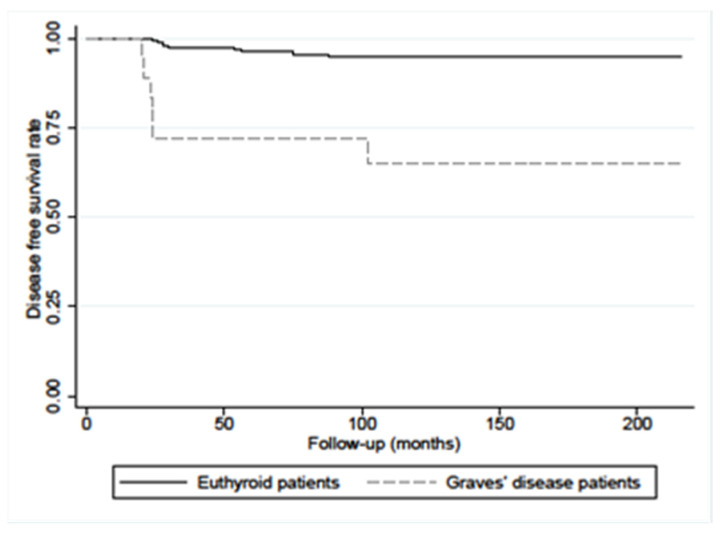
Disease-free survival (DFS) in PC/GD patients (n.18) and in matched euthyroid PC control patients (n.164) with microcarcinoma (≤1 cm) showed a statistical difference (*p* < 0.0001, calculated with log-rank test).

**Figure 5 diagnostics-12-02801-f005:**
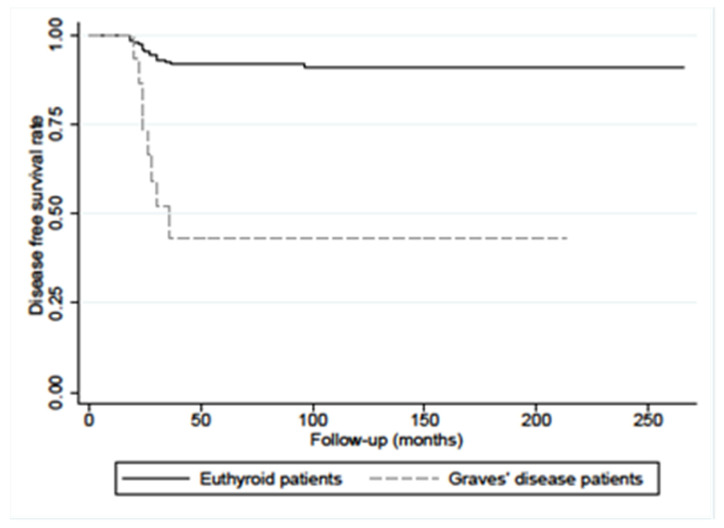
Disease-free survival (DFS) in PC/GD patients (n.15) and in matched euthyroid PC control patients (n.148) with carcinoma >1 cm showed a statistical difference (*p* < 0.0001 calculated with log-rank test).

**Table 1 diagnostics-12-02801-t001:** Demographic and histological characteristics of 33 patients with papillary carcinoma (PC) and associated Graves’ disease (GD), and 312 patients with PC without GD as controls at the surgery of primary carcinoma.

	PC Patients with GD (33 Cases)	PC Patients without GD as Controls (312 Cases)	*p*-Value
**Age (years)**	26 (78.8%) < 55; 7 (21.2%) ≥ 55	217 (69.5) < 55; 95 (30.5%) ≥ 55	0.269
**Age (years), mean ± SD**	45.1 ± 11.3 (median 45)	48.3 ± 9.1 (median 48)	0.0674
**Sex (F/M)**	18 (54.5%)/15 (45.5%)	216 (69.2%)/96 (30.8%)	0.09
**Histology**	28 (84.8%) PC classic variant 5 (15.2%) PC follicular variant	264 (84.6%) PC classic variant 48 (15.4%) PC follicular variant	0.972
**Tumor size (mm)**	18 (54.5%) ≤ 10/15 (45.5%) > 10	164 (52.6%) ≤ 10/148 (47.4%) > 10	0.828
**Tumor size (mm) mean ± SD**	10.85 ± 7.59	11.23 ± 8.90	0.813
**PC as incidental findings**	15 (45.5%)	31 (10%)	<0.001
**Multifocality**	6 (18.2%)	Absent	<0.001
**Minimal extrathyroid extension (mETE)**	3 (9%)	Absent	0.001
**Neck lymph node (LN) metastasis**			0.009
N0	31 (94%)	312 (100%)	
N1a	0	0	
N1b	2 (6%)	0 (0%)	
**Distant metastasis**	Absent	Absent	1
**Exophthalmos**	6 (18.2%)	Absent	<0.001
**TRAbs**	available in 15 (45.4%) cases, all positive	Not assayed	NA
**Risk stratification**			0.003
High risk (H)	2 (6%)	0	
Low risk (L)	19 (57.6%)	148 (47.4%)	
Very low risk (VL)	12 (36.4%)	164 (52.6%)	

**Table 2 diagnostics-12-02801-t002:** Clinical-pathological data of 33 patients with PC and associated GD at the surgery of primary tumor and at long-term follow-up. The carcinomas of Patients 1–14 (Group 1) had aggressive behavior, while Cases 15–33 (Group 2) did not develop metastases.

	At Surgery												At Follow-Up			
Patients n.	Sex	Age	Clinical Diagnosis	Histology	Size (mm)	Focality	ETE	Neck LN and Distant Metastasis	Risk Stratification	TNM	Exophthalmos (E)	TRAbs	Planar WBS (n. foci)	SPECT/CT (n. foci)	Tg (ng/mL)	AbTg (IU/mL)
** *GROUP 1* **																
**1**	F	27	DTH-ni	PC	7	unifocal			VL	T1aN0M0		Ar	0	1 LTC (oc)	und	absent
**2**	F	40	MNG-i	PC	5	unifocal			VL	T1aN0M0		Ar	0	1 SM (oc)	0.8	absent
**3**	M	45	DTH-i	PC	8	unifocal			VL	T1aN0M0	E	Pos	0	1 LTC (oc) + 1 SM (oc)	2.1	absent
**4**	F	46	DTH-i	PC	6	unifocal			VL	T1aN0M0		Ar	0	1 LTC (oc)	und	absent
**5**	M	48	MNG-i	PC	4	unifocal			VL	T1aN0M0		Ar	0	1 LTC (oc)	und	absent
**6**	M	45	DTH-ni	PC	37	unifocal			L	T2N0M0		Ar	2 residues + 1 residue (W)	2 residues + 1 PT (W)	und	absent
**7**	F	27	MNG-ni	PC	18	unifocal			L	T1bN0M0		Pos	0	2 LTC (oc)	und	absent
**8**	M	59	MNG-ni	PC	11	unifocal			L	T1bN0M0		Pos	2 unclear	2 LTC	1.9	absent
**9**	M	64	DTH-ni	PC	11	unifocal			L	T1bN0M0		Ar	0	1 SM (oc)	und	143
**10**	M	67	DTH-ni	PC/FV	11	unifocal			L	T1bN0M0		Pos	0	1 SM (oc)	und	absent
**11**	M	40	DTH-ni	PC	20	unifocal		Neck LN	H	T1bN1bM0	E	Pos	0	1 LTC (oc)	und	absent
**12**	M	51	MNG-ni	PC/FV	20	unifocal		Neck LN	H	T1bN1bM0		Ar	5 lung	5 lung	33	absent
**13**	F	45	MNG-i	PC	7	unifocal	mETE		L	T1aN0M0	E	Pos	0	1 SM (oc)	0.8	absent
**14**	M	37	MNG-ni	PC/FV	20/10	multifocal	mETE		L	T1bN0M0		Ar	0	1 LTC (oc)	1.2	absent
** *GROUP 2* **																
**15**	F	42	MNG-i	PC	2	unifocal			VL	T1aN0M0		Ar	0	0	und	absent
**16**	F	38	DTH-i	PC	3	unifocal			VL	T1aN0M0		Ar	0	0	und	absent
**17**	M	43	DTH-i	PC	4	unifocal			VL	T1aN0M0		Pos	0	0	und	absent
**18**	M	65	DTH-i	PC/FV	8	unifocal			VL	T1aN0M0	E	Pos	0	0	und	absent
**19**	F	44	DTH-i	PC	4	unifocal			VL	T1aN0M0		Pos	0	0	und	absent
**20**	F	54	DTH-i	PC	6	unifocal			VL	T1aN0M0		Ar	0	0	und	absent
**21**	M	59	MNG-i	PC	7	unifocal			VL	T1aN0M0	E	Pos	0	0	und	absent
**22**	M	38	DTH-ni	PC	20	unifocal			L	T1bN0M0		Ar	0	0	und	absent
**23**	F	50	DTH-ni	PC	25	unifocal			L	T1bN0M0		Pos	0	0	und	absent
**24**	F	21	DTH-ni	PC/FV	15	unifocal			L	T1bN0M0		Ar	0	0	und	absent
**25**	F	48	MNG-ni	PC	12	unifocal			L	T1bN0M0		Ar	0	1 residue	und	absent
**26**	F	38	MNG-ni	PC	11	unifocal			L	T1bN0M0		Ar	0	0	und	absent
**27**	F	58	DTH-ni	PC	11	unifocal			L	T1bN0M0		Ar	0	1 residue	und	absent
**28**	F	43	DTH-ni	PC	12	unifocal	mETE		L	T1bN0M0		Ar	0	0	und	absent
**29**	M	22	MNG-ni	PC	10/8	multicentric			L	T1aN0M0		Pos	0	2 residues	und	absent
**30**	F	46	MNG-i	PC	5/2.5	multicentric			L	T1aN0M0	E	Pos	0	0	und	absent
**31**	F	45	MNG-i	PC	2/2	multicentric			L	T1aN0M0		Pos	0	0	und	absent
**32**	M	55	MNG-i	PC	8/2	multicentric			L	T1aN0M0		Pos	0	2 residues	und	absent
**33**	F	40	MNG-ni	PC	8/2/2	multicentric			L	T1aN0M0		Ar	0	0	und	absent

DTH: diffuse thyroid hyperplasia; MNG: multinodular goiter; i: incidentaloma; ni: not incidentaloma; PC: papillary carcinoma; PC/FV: papillary carcinoma follicular variant; ETE: tumor extrathyroid extension; mETE: minimal tumor extrathyroid extension; LN: lymph node metastasis; H: high risk; L: low risk; VL: very low risk; TRAbs: thyroid-stimulating hormone receptor antibodies; Ar: absence of results; Pos: positive; LTC: laterocervical LN metastasis; SM: submandibular LN metastasis; PT: paratracheal LN metastasis; lung: lung metastasis; oc: occult; W: wrongly classified at WBS; Tg: thyroglobulin; AbTg: antithyroglobulin antibodies; und: undetectable.

**Table 3 diagnostics-12-02801-t003:** Clinical-pathological data of 21/312 patients with PC without GD at the surgery of the primary tumor and at long-term follow-up. All these patients without risk factors at surgery developed metastases after thyroidectomy and radioiodine ablation.

	At Surgery										At Follow-Up			
Patients n.	Sex	Age	Clinical Diagnosis	Histology	Size (mm)	Focality	ETE	Neck LN and Distant Metastasis	Risk Stratification	TNM	Planar WBS (n. foci)	SPECT/CT (n. foci)	Tg (ng/mL)	AbTg (IU/mL)
**1**	F	44	STN-ni	PC/FV	17	unifocal	absent	absent	L	T1bN0M0	1 unclear	1 PT	2.6	absent
**2**	F	34	STN-ni	PC	30	unifocal	absent	absent	L	T2N0M0	0	1 LTC (oc)	und	absent
**3**	F	47	MNG-ni	PC/FV	12	unifocal	absent	absent	L	T1bN0M0	0	1 LTC (oc)	und	absent
**4**	F	34	MNG-ni	PC	15	unifocal	absent	absent	L	T1bN0M0	1 unclear	1 LTC	6	absent
**5**	F	71	MNG-ni	PC	11	unifocal	absent	absent	L	T1bN0M0	0	1 PT (oc)	0.9	absent
**6**	F	40	STN-ni	PC	20	unifocal	absent	absent	L	T1bN0M0	0	1 PT (oc)	und	absent
**7**	M	54	STN-ni	PC	11	unifocal	absent	absent	L	T1bN0M0	0	1 PT (oc)	1.1	absent
**8**	F	53	MNG-i	PC	5	unifocal	absent	absent	VL	T1aN0M0	1 unclear	1 PT	13	absent
**9**	F	28	MNG-i	PC	5	unifocal	absent	absent	VL	T1aN0M0	0	1 SM (oc)	2.3	absent
**10**	F	44	MNG-i	PC	5	unifocal	absent	absent	VL	T1aN0M0	0	1 SM (oc)	3.2	absent
**11**	F	28	MNG-i	PC	4	unifocal	absent	absent	VL	T1aN0M0	1 unclear	1 SM	2.6	absent
**12**	F	34	STN-ni	PC	12	unifocal	absent	absent	L	T1bN0M0	0	1 LTC (oc)	3	absent
**13**	F	42	MNG-ni	PC/FV	12	unifocal	absent	absent	L	T1bN0M0	1 unclear	1 LTC	und	absent
**14**	F	38	STN-ni	PC/FV	15	unifocal	absent	absent	L	T1bN0M0	1 unclear	1 LTC	und	absent
**15**	F	46	MNG-i	PC	5	unifocal	absent	absent	VL	T1aN0M0	0	2 LTC (oc) + 1 SC (oc)	und	absent
**16**	F	64	MNG-i	PC	2	unifocal	absent	absent	VL	T1aN0M0	2 residues	2 residues + 1 SM (oc)	und	absent
**17**	F	52	MNG-i	PC/FV	2	unifocal	absent	absent	VL	T1aN0M0	1 residue + 1unclear	1 residue + 1 LTC	6	absent
**18**	M	21	STN-ni	PC	20	unifocal	absent	absent	L	T1bN0M0	2 unclear	2 M	9.6	absent
**19**	F	60	STN-ni	PC/FV	15	unifocal	absent	absent	L	T1bN0M0	2 unclear	1 LTC + 1 PV	4.2	absent
**20**	F	61	MNG-ni	PC	11	unifocal	absent	absent	L	T1bN0M0	1 unclear	1 SC + 1 SM (oc)	und	absent
**21**	F	62	MNG-ni	PC	5	unifocal	absent	absent	VL	T1aN0M0	1 unclear	1 LTC + 1 PT (oc)	6.5	absent

STN: single nodule; MNG: multinodular goiter; i: incidentaloma; ni: not incidentaloma; PC: papillary carcinoma; PC/FV: papillary carcinoma follicular variant; ETE: tumor extrathyroid extension; LN: lymph node metastasis; L: low risk; VL: very low risk; LTC: laterocervical LN metastasis; SM: submandibular LN metastasis; PT: paratracheal LN metastasis; SC: supraclavicular LN metastasis; M: mediastinum LN metastasis; PV: pelvic LN metastasis; oc: occult; Tg: thyroglobulin; AbTg: antithyroglobulin antibodies; und: undetectable.

**Table 4 diagnostics-12-02801-t004:** The multivariate logistic regression shows that PC/GD patients had a 10-fold higher risk of developing metastasis than PC patients without GD (*p* < 0.001). The multivariate analysis includes independent variables resulting with a *p* < 0.10 in the univariate analysis.

Variables	OR (95% CI)	*p*-Value	aOR (95% CI)	*p*-Value
**Age (years)**	0.97 (0.94–1.01)	0.122		
**Age < 55 years**	1.47 (0.64–3.35)	0.361		
**Male gender**	1.60 (0.75–3.43)	0.226		
**PC follicular variant**	2.09 (0.92–4.76)	0.078	2.31 (0.94–5.67)	0.067
**Tumor size (mm)**	1.01 (0.97–1.05)	0.547		
**Microcarcinoma (≤10 mm)**	0.56 (0.28–1.15)	0.114		
**Multifocality**	1.79 (0.20–15.80)	0.599		
**Minimal extrathyroid extension (mETE)**	18.72 (1.65–212.1)	0.018	2.62 (0.20–33.53)	0.459
**Neck lymph node (LN) metastasis**	1			
**Graves’ disease**	10.2 (4.50–23.19)	<0.001	9.79 (4.10–23.35)	<0.001

## Data Availability

The data presented in this study are available on reasonable request from the corresponding author.

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
