# Peer review of "A Comparative Follow-Up Study of Patients with Papillary Thyroid Carcinoma Associated or Not with Graves’ Disease"

_diagnostics, 2022, doi:10.3390/diagnostics12112801_

Round 1
Reviewer 1 Report
The topic of the influence of coexisting GD on the course of PTC is quite interesting for the practitioners involved in the management of thyroid cancer. However, the manuscript needs substantial improvements (including repeated data analysis) before being suitable for publishing.
First of all, the manuscript needs linguistic editing badly, as some of its parts are hard to comprehend (some mistakes present in the first sentence of the paper: in particularly instead of particularly). It would make it easier to review the paper if the lines were numbered.
Page 1. Introduction line 3. The authors conclude that GD is due to the activation of TRAbs, whereas GD is due to the activation of the TSH receptor by TRAbs, and the cell proliferation (which, hypothetically, might promote carcinogenesis) is due to that activation rather than thyroid hormone excess. Otherwise, TSH suppression with L-thyroxin would not be applied in DTC management.
Page 2, line 6. It should be explained why IGFs might be involved in thyroid cancer formation in GD.
Page 4, 4th paragraph. As the topic of the paper is not related to the superiority of SPECT/CT over the WBS - that part of the introduction should be omitted.
Material and methods:
It would be worth mentioning when the thyroidectomies were performed. The % of central and lateral neck dissections should be added instead of "mostly accompanied", etc.
Page 3, line 4. What does it mean that the treatment with methimazole was inconclusive?
Page 9, line 9. The authors have stratified the patients into age groups according to AJCC 8th ed. However, the meaning of the sentence in its current form is different (linguistic editing!).
Page 9, line 25. It needs to be explained why the authors have chosen the European TThyroid Cancer Taskforce classification, as the ATA risk stratification system is currently most frequently applied.
Table 1. It would be helpful to add the percentage to the numbers and show in the table if the difference between groups was statistically significant (if applicable). The median age of the patients should also be included. It is not clear how the control group was established (every patient undergoing surgery during the same period? 5 control, age and sex-matched patients to every patient in the GD group?). Was there any difference in RAI treatment schedule between the groups (L-thyroxin withdrawal vs. hrTSH stimulation)?
Some patients underwent the WBS while on L-thyroxin withdrawal, others under hrTSH stimulation - why? Was there a difference in stimulation mode between the GD and control groups? Which assays were used to measure Tg and Ab-Tg? There given cut-offs are the cut-offs for "Tg-positivity" or the assay detection limit? And for stimulated or suppressed Tg (if recognised as positive).
It would be good to know how many of the control group patients were diagnosed with autoimmune thyroid disease (based on pathology reports or Ab-Tg).
It seems that RAI-negative metastases were not considered (see page 5, 2nd paragraph). Is that true?
Logistic regression results may be presented in a table. A multiple regression model should also be applied (to test for confounders) to assess the difference between groups.
Page 8, line 11. TKI treatment cannot be defined as immunotherapy.
Page 12, 2nd and 5th paragraphs. it should be specified to which subgroup the sentences relate.
As the TRAbs were available only for less than half of GD patients, it seems prudent to omit any conclusions/results related to them.
Author Response
Dear Reviewer,
attached here there is a file with replies to your comments.

Reviewer 2 Report
Overall, the article is interesting, statistics seems acceptable.
Unfortunately, English style and writing are severely inadequate, and this compromise the reading and clarity of the whole paper. This article needs to be revised by a mother tongue scientist.
Here some minor comments:
Title is redundant and confusing; it should be shortened.
Introduction: improve English writing, the long digression on WBS/SPECT is irrelevant to the topic.
Methods: improve English writing. Some results (descriptive statistics) are reported in the methods. Table 1: missing p values comparison of basal characteristics between patients with/without GD. Again, deep details on WBS/SPECT are irrelevant to the topic.
Statistics: ok
Results: Improve English writing and consistency (e.g. sometimes Authors write 10/20 other times ten/20). Add some details in the GD cases: How was the control of hyperthyroidism in GD patients? They were all patients with active disease? Did you collect disease duration? How many of them were on thiamazole?...
In Kaplan Meyer, the distinction Euthyroid/Graves’ Disease is misleading… could be better differentiate patients with/without PTC.
Discussion: Comment article 10.1007/s40618-020-01232-6, in this series the presence of germinal centers perfectly predicted not having DTC in GD patients, overall suggesting a protective role of a florid inflammatory infiltrate onto carcinogenesis --> had the patients with GD and aggressive PTC an active disease or it was cured?
Author Response
Dear Reviewer,
attached here there are the replies to your comments.

Round 2
Reviewer 1 Report
Please find the attachment.

Author Response
Dear reviewer,
attached here, you'll find the replies to your comments.
Kind regards
Angela Spanu
